# Immunotherapy-induced antibodies to endogenous retroviral envelope glycoprotein confer tumor protection in mice

**Byong H. Kang**[1,2], **Noor Momin**[1,2], **Kelly D. Moynihan**[1,2,3], **Murillo Silva**[1], **Yingzhong Li**[1,2], **Darrell J. Irvine**[1,2,3,4,5], **K. Dane Wittrup**[1,2,6]*

**1** Koch Institute for Integrative Cancer Research, Massachusetts Institute of Technology, Cambridge, Massachusetts, United States of America, **2** Department of Biological Engineering, Massachusetts Institute of Technology, Cambridge, Massachusetts, United States of America, **3** Ragon Institute of Massachusetts General Hospital, Massachusetts Institute of Technology and Harvard University, Cambridge, Massachusetts, United States of America, **4** Howard Hughes Medical Institute, Chevy Chase, Maryland, United States of America, **5** Department of Materials Science and Engineering, Massachusetts Institute of Technology, Cambridge, Massachusetts, United States of America, **6** Department of Chemical Engineering, Massachusetts Institute of Technology, Cambridge, Massachusetts, United States of America

* wittrup@mit.edu

**Data Availability Statement:** All data are within the paper and its Supporting Information files.

**Funding:** This work was supported in part by the Ragon Institute of MGH, MIT and Harvard, the

## Abstract

Following curative immunotherapy of B16F10 tumors, ~60% of mice develop a strong antibody response against cell-surface tumor antigens. Their antisera confer prophylactic protection against intravenous challenge with B16F10 cells, and also cross-react with syngeneic and allogeneic tumor cell lines MC38, EL.4, 4T1, and CT26. We identified the envelope glycoprotein (env) of a murine endogenous retrovirus (ERV) as the antigen accounting for the majority of this humoral response. A systemically administered anti-env monoclonal antibody cloned from such a response protects against tumor challenge, and prophylactic vaccination against the env protein protects a majority of naive mice from tumor establishment following subcutaneous inoculation with B16F10 cells. These results suggest the potential for effective prophylactic vaccination against analogous HERV-K env expressed in numerous human cancers.

## Introduction

Success of checkpoint blockade therapies in the past decade has clearly demonstrated the dominant role T cells play in an antitumor response and immunosurveillance [1, 2]. However, the functional role of B cells and antibodies (Abs) in cancer remains less clear [3, 4]. High expression levels of B cell signature genes correlated with improved survival in patients with melanoma, lung adenocarcinoma, pancreatic adenocarcinoma, and head and neck squamous cell carcinoma, but with poor prognosis in patients with glioblastoma and clear cell renal cell carcinoma [5–7]. In some mouse models, B cells have been shown to exert pro-tumor effects by promoting metastasis [8], angiogenesis [9], and contributing to an immunosuppressive tumor microenvironment [10–12]. In most cases, however, they have been shown to be critical in

Koch Institute Frontier Research Program, the
Kathy and Curt Marble Cancer Research Fund, and
the Mayo Clinic – Koch Institute Cancer Solutions
Team Grant funding, and the National Cancer
Institute, CA174795.

**Competing interests:** The authors have declared
that no competing interests exist.

supporting antitumor effects by augmenting T-cell responses [13–17] and by producing anti-tumor Abs that can induce antibody-dependent cellular cytotoxicity (ADCC) [18] and enhance antigen presentation by dendritic cells [17].

To better understand effective humoral responses to cancer, we investigated anti-tumor Abs induced by curative immunotherapies in mouse models. Successful immunotherapies in mice mediate primary tumor rejection and exhibit protection from a secondary challenge, signifying successful formation of immunological memory [19–21]. This protection has been shown to be mediated by not only cellular immune response but also humoral response against tumor cells [20, 22]. We have previously reported that immunotherapy-induced antibodies (iiAbs) from cured mice are able to recognize and bind multiple antigens on cognate tumor cells. Additionally, antisera from cured mice protect naive mice from an intravenous tumor challenge, indicating that the iiAbs alone are sufficiently protective *in vivo* [20]. Here, we observe that these iiAbs can recognize and delay the growth of heterologous tumor cells, strongly motivating the identification of their cognate antigens. We report that two of the recognized antigens are products of endogenous ecotropic murine leukemia virus (eMLV), with the envelope glycoprotein (env) of eMLV as the dominant cell-surface antigen targeted by the iiAbs in cured mice. Through the use of an anti-env monoclonal antibody (mAb) and subunit vaccination, we show that the Ab response against eMLV env confers tumor protection.

## Materials and methods

### Mice

C57BL/6NTac mice (6–9 weeks old) were purchased from Taconic. All animal work was conducted under the approval of the Massachusetts Institute of Technology (MIT) Division of Comparative Medicine in accordance with federal, state, and local guidelines.

### Cells

B16F10, CT26, EL.4, EMT6, and 4T1 cells were purchased from ATCC American Type Culture Collection (ATCC). KP 2677 cells were a gift from T. Jacks (MIT). MC38 cells were a gift from J. Schlom (National Cancer Institute). TC-1 cells were a gift from T.C. Wu (Johns Hopkins University). B16F10, EL.4, EMT6, KP2677, and MC38 cells were cultured in Dulbecco's modified Eagle's medium (ATCC) supplemented with 10% heat-inactivated fetal bovine serum (HI FBS; Life Technologies), 100 units/ml penicillin, and 100 μg/ml streptomycin (Life Technologies). CT26, TC-1, and 4T1 cells were cultured in RPMI (ATCC) supplemented with 10% HI FBS, 100 units/ml penicillin, and 100 μg/ml streptomycin. All tumor cell lines were cultured at 37°C and 5% $CO_2$. HEK-293F cells were purchased from Life Technologies and cultured in FreeStyle 293 Expression Medium (Life Technologies) at 37°C and 8% $CO_2$.

### Tumor inoculation and treatments

Mice were anesthetized with 3% isoflurane in 100% oxygen at 0.5 L/min for tumor inoculation and treatments. For initial treatment with the AIPV regimen, an inoculum of $10^6$ B16F10 cells was injected s.c. on the flank of mice in 50 μl sterile PBS. Mice were administered TA99 (100 μg; i.p.), MSA-IL2 (30 μg; i.p.), anti-PD-1 (RMP1-14, BioXCell, 200 μg, i.p.), amph-CpG (1.24 nmol; s.c.), and amph-peptide (20 μg, s.c.) on days 8, 15, 22, 29, and 36. 4

For the heterologous challenge study, mice were first given a secondary challenge of $10^5$ B16F10 cells inoculated on the opposite flank on week 12. On week 24, an inoculum of $10^5$ MC38 cells was injected s.c. on the back of mice in 50 μl sterile PBS.

For the prophylactic monotherapy study, mice were administered TA99 or 1E4.2.1 (200 μg; i.p.) on days –1, 2, 5, 8, and 11. An inoculum of $10^5$ B16F10 cells was injected s.c. on the flank of mice in 50 μl sterile PBS on day 0.

Tumor size was measured as an area (longest dimension × perpendicular dimension) every 2–3 days, and mice were euthanized with 100% CO2 at 3 L/min followed by cervical dislocation when tumor area exceeded 100 mm$^2$. No unexpected mortality or adverse events were observed.

## Flow cytometry

Abs to mouse IgG1 (RMG1-1), IgG2a/c (RMG2a-62), IgG2b (RMG2b-1), IgG3 (RMG3-1) were purchased from BioLegend and labeled with Alexa Fluor 647 NHS Ester (Life Technologies). For labeling mouse cell lines, adherent cells were dissociated from flasks with CellStripper (Corning), washed in PBS (Corning) with 1% BSA (Sigma; PBSA), incubated with 1% mouse sera or 10 μg/ml Ab for 1 h on ice with or without the presence of 50 μg/ml soluble monomeric gp70 or gag, then with 1 μg/ml secondary Ab for 30 min on ice. Viability was assessed by PI or DAPI staining. Cells were analyzed using BD FACS LSR II, BD FACSCanto, and BD Accuri flow cytometers, and data were analyzed using FlowJo.

## Membrane protein extraction

Membrane proteins of B16F10 and MC38 cells were extracted with Plasma Membrane Protein Extraction Kit (Abcam) according to the manufacturer's instructions. The membrane protein pellet was solubilized in Pierce IP Lysis Buffer (Thermo Fisher) with 1x cOmplete Protease Inhibitor (Roche) for 1D immunoblotting and in 7M urea (Sigma), 2M thiourea (Sigma), 2% CHAPS (Sigma), 50 mM DTT (Sigma), and 1% ZOOM Carrier Ampholytes (Thermo Fisher) for 2D immunoblotting.

## Serum immunoblot

Sera were collected in MiniCollect Serum Separator tubes (Greiner Bio-One). Abs to mouse IgG2a/c and IgG2b were labeled with IRDye 800CW NHS Ester (LI-COR). Twenty μg/well of membrane proteins were run on reducing NuPAGE 4–12% Bis-Tris gels (Life Technologies) in MES buffer and transferred to nitrocellulose membranes with iBlot (Life Technologies). After blocking with blocking buffer (0.5x Odyssey Blocking Buffer in TBS; LI-COR) for 1 h at room temperature, membranes were incubated with serum diluted 1:1000 in blocking buffer overnight at 4˚C. Membranes were probed with IRDye-800CW-conjugated anti-mouse IgG Abs diluted 1:2500 in blocking buffer for 1 h at room temperature and imaged on a LI-COR Odyssey Infrared Imaging System.

## 2D immunoblot

Standard proteins BSA, soybean trypsin inhibitor, and equine myoglobin (Sigma) were biotinylated with NHS-LC-Biotin (Life Technologies). Each ZOOM IPG Strip pH 3–10NL or pH 4.5–5.5 (Life Technologies) was hydrated with 100 μg membrane proteins with or without 0.5 μg of each standard protein in a ZOOM IPGRunner Cassette (Life Technologies) overnight at room temperature. ZOOM IPGRunner Mini-Cell (Life Technologies) was assembled according to the manufacturer's instructions and isoelectric focusing was performed at 175 V for 15 min, 175–2000 V ramp for 45 min, and 2000 V for 1 h. IPG strips were equilibrated with 1X NuPAGE LDS Sample Buffer with 50 mM DTT for 15 min, then with 1X NuPAGE LDS Sample Buffer with 125 mM iodoacetamide for 15 min. Equilibrated IPG strips were run on

NuPAGE 4–12% Bis-Tris ZOOM gels in MES buffer. Gels were transferred to nitrocellulose membranes with iBlot at 20 V for 2 min 20 s for partial transfer of ~25 kDa proteins and for 3 min or 3 min 30 s for ~80 kDa proteins. Membranes were probed with IRDye-800CW-conjugated anti-mouse IgG2b and IgG2c and Alexa-Fluor-647-conjugated streptavidin for fluorescent imaging. For chromogenic immunoblot, membranes were probed with anti-mouse IgG2b and IgG2c Abs biotinylated with NHS-LC-Biotin (Life Technologies), then with HRP-conjugated streptavidin (R&D Systems), and developed with 1-Step TMB-Blotting Substrate Solution (Thermo Fisher).

### Silver stain and mass spectrometry

Gels were silver stained with Pierce Silver Stain for Mass Spectrometry (Thermo Fisher) according to the manufacturer's instructions, except that they were incubated in silver stain for 15 minutes. Silver stained gels were overlaid on chromogenic immunoblots to identify and excise the spots of interest on the gels. Excised gels were digested with trypsin into polypeptide fragments and analyzed by liquid chromatography/tandem mass spectrometry (LC/MS/MS).

### Protein and vaccine production

Therapeutic proteins and vaccines were produced as described [20]. Total RNA was isolated from B16F10 cells with RNeasy Mini Kit (Qiagen) and reverse transcribed with SuperScript III RT and random primers (Invitrogen) to synthesize B16F10 first-strand cDNA. eMLV env and gag sequences were amplified from B16F10 first-strand cDNA. eMLV env monomeric gp70 with a C-terminal His tag was cloned into the gWIZ vector (Gelantis) by Gibson assembly and produced by transiently transfecting HEK293F cells with the plasmid and polyethylenimine. The RBD was cloned by site-directed mutagenesis (NEB) and produced as gp70. Gag was expressed as a SUMO fusion using the pE-SUMOpro vector (LifeSensors) in Rosetta-gami 2 (DE) competent cells. Heavy and light chains of anti-env Abs were separately cloned into the gWIZ vector and anti-env Abs were produced by transiently co-transfecting HEK293F cells as above. His-tagged proteins were purified using TALON Metal Affinity Resin (Takara) and Abs were purified using rProtein A Sepharose Fast Flow resin (GE Healthcare). Endotoxin levels were below 0.1 total EU/dose as measured by the QCL-1000 chromogenic LAL assay (Lonza).

### Single B cell sorting

Spleen, inguinal and axillary lymph nodes, and femoral bone marrow were harvested from a mouse with anti-env Abs and single-cell dissociated as described [23]. Cells were labeled with fluorescent anti-B220, anti-mouse IgG2b, anti-mouse IgG2a/c, and monomeric gp70. Viable cells positive for B220, IgG2b or IgG2c, and gp70 were analyzed on BD FACSAria II and single-cell sorted into 96-well PCR plates containing lysis buffer with RNasin Plus RNAse inhibitor (Promega). After reverse transcription, heavy and light variable regions were amplified by semi-nested PCR as described [24].

### Cryo-fluorescence tomography

1E4 was labeled with Alexa Fluor 750 NHS Ester (Life Technologies) and 500 μg was intravenously injected into a naive mouse bearing a B16F10 tumor. Forty-eight hours later, the mouse was euthanized and fast frozen in hexane with dry ice. Coronal cryosectioning and white-light imaging were performed by EMIT using a Xerra imager. Following each sequential removal of 50 μm thick slices, the tissue-embedded block was imaged at 30 μm in-plane

resolution. A 3D image volume of the mouse was generated through multiplanar reformation using 3D Slicer software for anatomic visualization.

## Generation of the *env*-knockdown B16F10

sgRNA candidates against eMLV env were designed with GPP sgRNA Designer (Broad Institute) and 5 were cloned into the pSpCas9(BB)-2A-GFP vector. B16F10 cells were seeded to ~40% confluency a day before were transiently transfected with the plasmid and Lipofectamine 2000 (Life Technologies). Two days after the transfection, the cells were dissociated and live GFP+ cells were sorted on BD FACSAria II and expanded. Upon expansion, the cells were labeled with Alexa-Fluor-647-conjugated 1E4 or 1E4.2.1 to sort negative populations. Negative selection was performed 3 more times to enrich the population negative for the cell-surface env expression. B16F10 cells knocked down of env with the sgRNA sequence 5′-TGGAGACCGAGAAACGGTGT-3′ were used for flow cytometry experiments.

## scFv affinity-maturation by yeast surface display

Variable regions of 1E4 heavy and light chains were cloned as an scFv into the pCTCON2 vector and expressed on yeast. Yeast cells were incubated with biotinylated monomeric gp70 and chicken anti-c-myc Ab (Exalpha), washed, stained with Alexa-Fluor-488-conjugated streptavidin and Alexa-Fluor-647-conjugated anti-chicken IgG secondary Ab (Invitrogen), and analyzed by flow cytometry. The affinity of 1E4 scFv was improved by two rounds of affinity maturation as previously described [25–27], with each round consisting of error-prone PCR and two to three rounds enrichment by FACS.

## Prophylactic vaccine study

ISCOM-like saponin adjuvant incorporating MPLA was prepared as described [28]. Naive mice were vaccinated s.c. at the tail base with 10 μg RBD adjuvanted with 5 μg adjuvant on weeks 0, 3, 6, 9, and 12. Two to three weeks after each vaccination, sera were collected to analyze Ab responses to gp70. An inoculum of $10^5$ B16F10 cells was injected s.c. on the flank of mice in 50 μl sterile PBS on week 18.

## Statistical analysis

Sample sizes were chosen based on previous experiments such that appropriate statistical tests could yield significant results. Statistical analysis was performed using the GraphPad Prism software. Kaplan-Meier survival curves were compared by log-rank Mantel-Cox test.

# Results

## iiAbs in cured mice recognize heterologous tumor cell lines

We have previously observed that the combination immunotherapy termed AIPV (short for anti-tumor-associated-antigen [anti-TAA] **A**b, extended-half-life **i**nterleukin-2, anti-**P**D-1 Ab, and amphiphile **v**accine) causes robust tumor rejection and immunological memory in several mouse models resulting in successful rejection of a secondary subcutaneous rechallenge. This combination therapy induced antisera that were able to recognize cognate tumor cells and protect naive mice from intravenous B16F10 challenge [20]. Surprisingly, we observed that the iiAbs of IgG isotype in ~60% mice cured of B16F10 were able to bind not only cognate tumor cells, but also heterologous tumor cells, including syngeneic tumor cells (Fig 1A) of C57BL/6 background and allogeneic tumor cells of

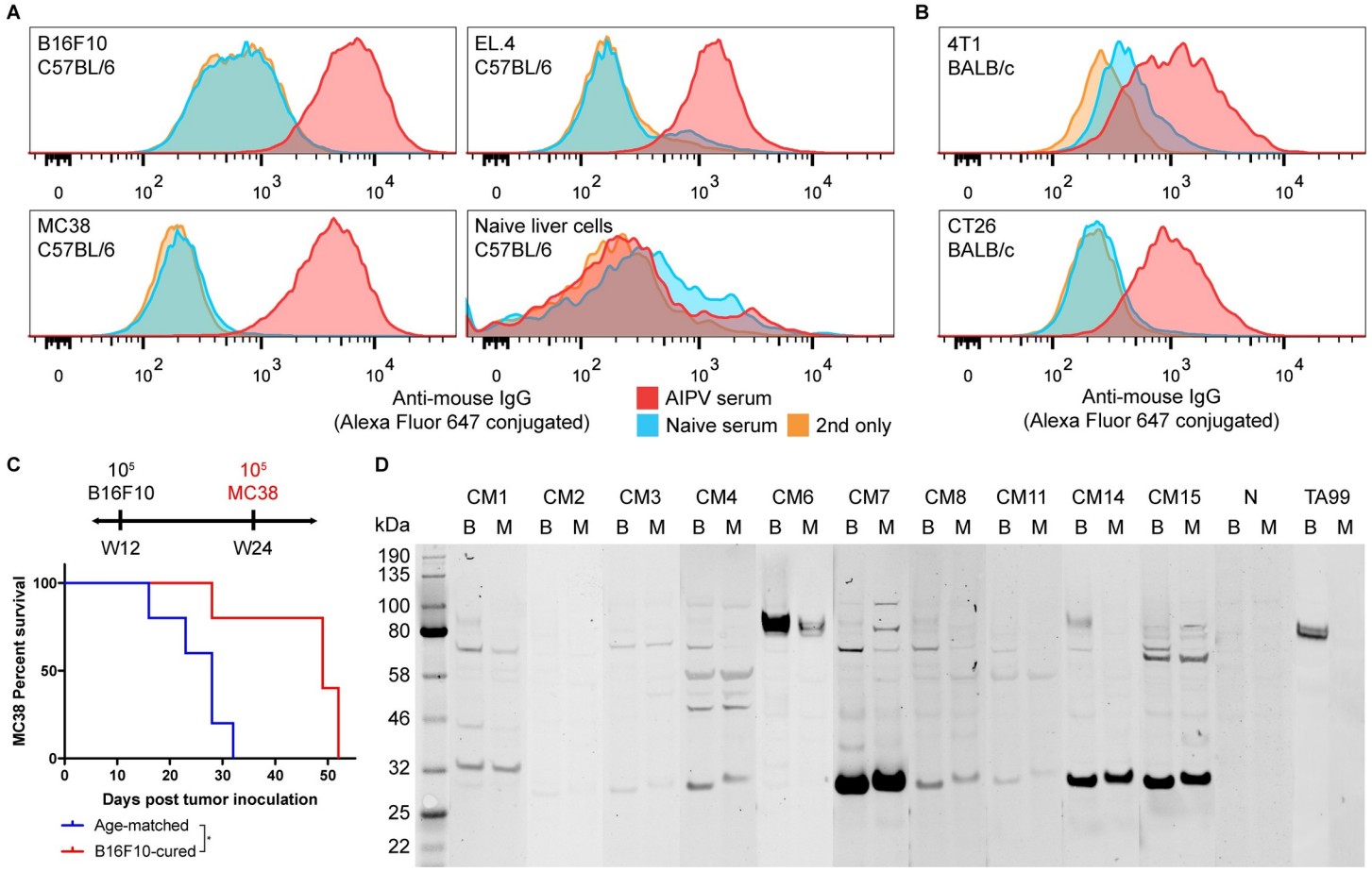

**Fig 1. iiAbs in cured mice cross-react with heterologous mouse tumor cell lines.** The indicated (A) syngeneic and (B) allogeneic tumor cells were incubated with 1% serum of AIPV-treated mice that had previous rejected B16F10 or naive mice. The cells were washed, stained with Alexa-Fluor-647-conjugated anti-mouse IgG secondary Ab, and analyzed by flow cytometry. (C) Age-matched mice (n = 5) and mice that were cured of B16F10 and rejected secondary challenge (n = 5) were inoculated with $10^5$ MC38 cells and left untreated. (D) B16F10 (B) and MC38 (M) membrane proteins were immunoblotted with AIPV serum (CM# for each cured mouse), naive serum (N), or TA99 and detected with IRDye-800CW-conjugated anti-mouse IgG2b and IgG2c Abs. *P < 0.05.

BALB/c background (Fig 1B). We observed that these class-switched Abs were predominantly of IgG2b and IgG2c isotypes, which are the most activating isotypes in C57BL/6 mice [29] (S1 Fig). We tested whether this broad cell-line specificity results in the control of heterologous tumors by inoculating MC38 in B16F10-cured mice that had rejected a secondary subcutaneous challenge. While we did not observe complete rejection, there was a significant growth delay compared to age-matched controls (Fig 1C). To better understand the specificities of these iiAbs, membrane proteins were isolated from B16F10 and MC38 cells and immunoblotted with the sera from mice that rejected a secondary challenge (Fig 1D). The presence of multiple detected antigens distinctive from TRP1, the antigen for anti-TAA mAb TA99 used in the therapy, indicated antigen spreading had occurred during therapy. However, the shared specificities between different cured mice, as well as between B16F10 and MC38 cells, suggested that the convergence of secondary epitopes to a handful of targets had also occurred. We sought to identify the targets of immunological memory as potentially exploitable universal tumor-agnostic antigens for novel immunotherapy.

## eMLV env and gag are targeted by the iiAbs

We took advantage of the strong signal over background observed on immunoblots and developed a 2D-gel-based technique to identify the targets of the iiAbs (Fig 2A). Two-dimensional gels have been a powerful tool in maximizing separation between different proteins though its recent application has been limited to 2D DIGE [30] due to inherent gel-to-gel variability. Using this technique, we first ran B16F10 or MC38 membrane proteins (with biotinylated standard proteins for 2D gels with a large pH gradient) on a 2D gel to further separate potential targets (Fig 2B). We empirically optimized the transfer conditions for different molecular weight proteins, identifying a transfer time and voltage for keeping proteins roughly equal on both blotting membrane and the gel. Then, the remaining gel was silver-stained to maximize the signal, and the membrane was chromogenically immunoblotted with serum. Using the standard proteins as landmarks, we overlaid the silver-stained gel over the chromogenic

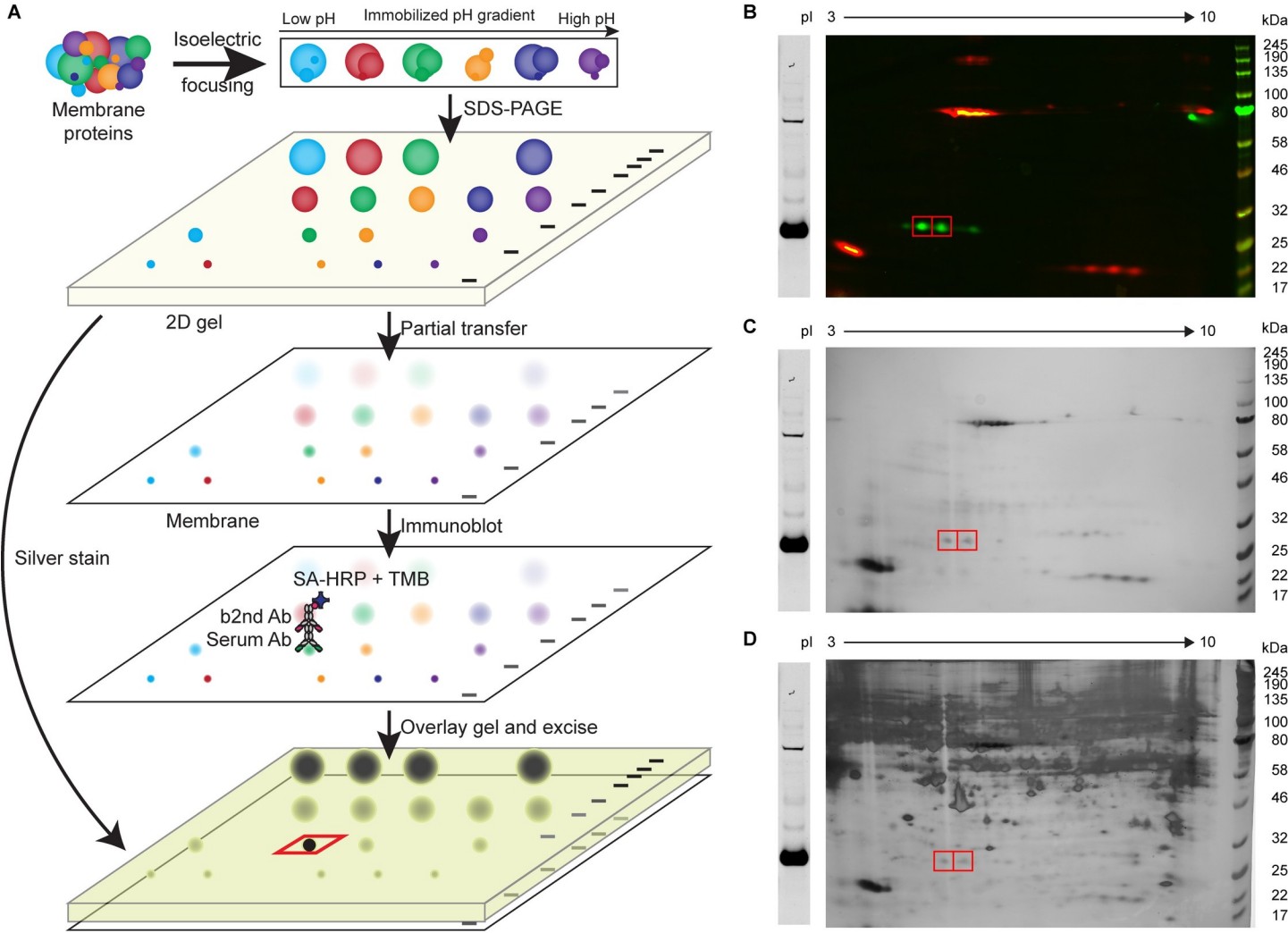

**Fig 2. Modified 2D immunoblotting allows for the identification of tumor antigens.** (A) The schematic describes the workflow to partially transfer a 2D gel and overlay silver-stained gel on chromogenic immunoblot to identify the spots of interest. (B) B16F10 membrane proteins and biotinylated BSA, soybean trypsin inhibitor, and equine hemoglobin were immunoblotted with CM7 serum and detected with IRDye-800CW-conjugated anti-mouse IgG2b and IgG2c Abs (green) and Alexa-Fluor-647-conjugated streptavidin (red). (C) Identical sample in (B) was partially transferred and chromogenically detected with biotinylated anti-mouse IgG2b and IgG2c Abs and HRP-conjugated streptavidin. (D) The gel remaining after the partial transfer was silver-stained and overlaid on the chromogenic immunoblot. Red boxes indicate the spots of interest.

immunoblot to locate the spots of interests on the gel for identification by mass spectrometry (Fig 2C and 2D). Since the signal is lowered due to separation in another dimension, we sought to identify the targets of two sera that showed the strongest signal on 1D immunoblots (Fig 1D). For each serum, its respective target existed in multiple isoforms with different isoelectric points, such that multiple spots could be excised to identify peptides in common (S2 and S3 Figs). These targets were identified by mass spectrometry to be env and gag of the endogenous ecotropic murine leukemia virus (eMLV), an endogenous retrovirus (ERV) found in C57BL/6 mice. It should be noted that this identification required searching for peptide sequences in the Uniprot or full RefSeq database without designating any species, as RefSeq database for *Mus musculus* does not include endogenous retroviral sequences (S1 and S2 Tables). Aware-ness of this potential blind spot in database annotation could be critical in the burgeoning field of identifying alternative tumor-specific antigens [31]. The extracellular domain of env (gp70 or SU) and gag were produced recombinantly and immunoblotting with corresponding serum confirmed their specificity (Fig 3A).

## eMLV env is the cell-surface antigen targeted by the iiAbs

We next sought to determine whether env or gag are cell-surface antigens on the tumor cells. We first tested whether soluble monomeric gp70 or gag could compete against the iiAbs

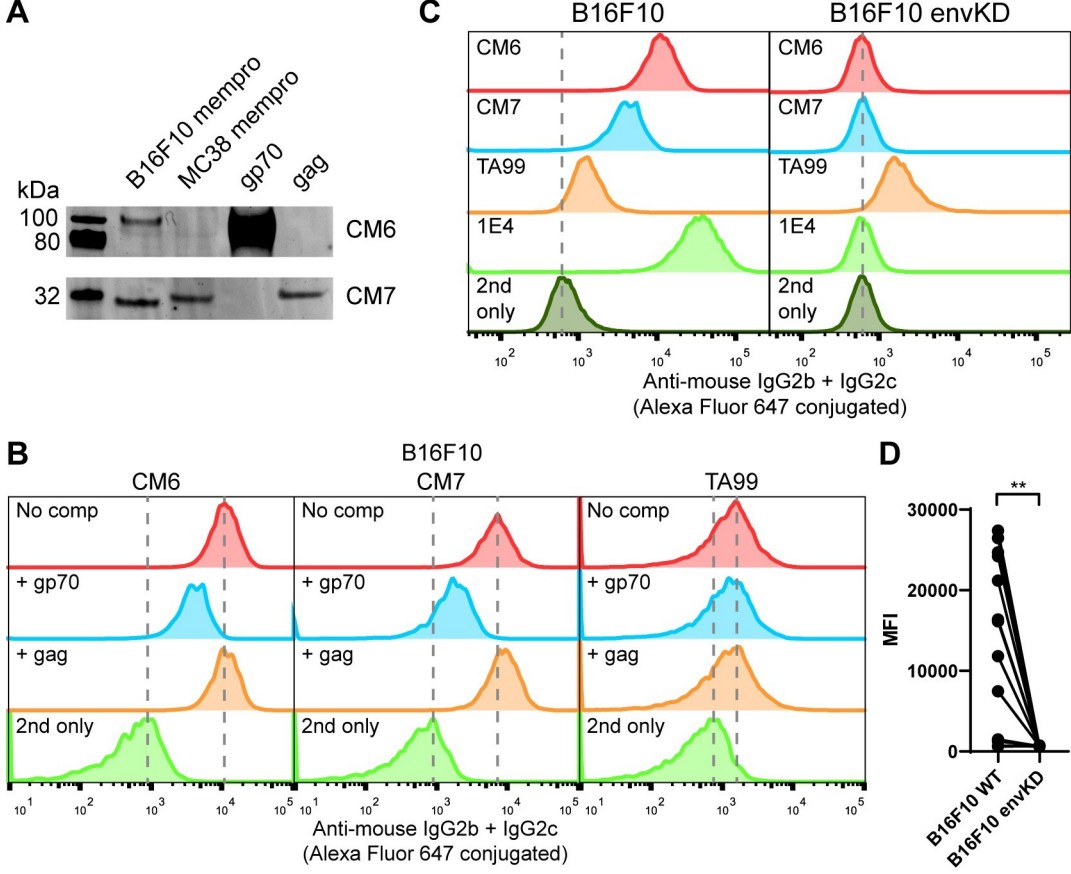

**Fig 3. eMLV gp70 is the dominant cell-surface antigen.** (A) eMLV gp70 and gag were recombinantly expressed and immunoblotted with CM6 and CM7 sera and detected as in Fig 1D. (B) B16F10 cells were incubated with 1% serum or 10 μg/ml TA99 in the presence of 50 μg/ml gp70 or gag and analyzed as in Fig 1A. (C) B16F10 or B16F10 envKO cells were incubated with 1% serum or 10 μg/ml TA99 or 1E4 and analyzed as in Fig 1A.

binding B16F10 cells. Gp70, but not gag, showed marked competition for antiserum binding by flow cytometry (Fig 3B). In fact, all of the sera with reactivity to B16F10 cells were competed by recombinant gp70 (S4 Fig), indicating that env is an immunodominant cell-surface antigen. We also tested sera of mice that were cured of B16F10 with a different therapeutic regimen (TA99 and four different constructs of intratumorally injected MSA-IL2) and found that all of the sera showing reactivity to B16F10 were competed by gp70 (S5A Fig). Additionally, we tested sera of mice cured of CT26 by intratumoral treatment with self-replicating RNAs encoding IL-12 [32] and those that showed cross-reactivity to B16F10 were also competed by gp70 (S5B Fig). This indicated that an anti-env Ab response is not unique to the AIPV therapy, and may likely be induced in other curative immunotherapies. We further tested whether the receptor-binding domain (RBD) of gp70 could compete against the iiAbs binding B16F10 cells. We found that the RBD could compete similarly to gp70, indicating that the RBD portion of gp70 contains predominant epitopes (S6 Fig). To conclusively determine whether there are other cell-surface antigens contributing to the reactivity of the iiAbs, we first sought to isolate anti-env mAbs. We harvested primary and secondary lymphoid organs from an AIPV-treated mouse and single-cell sorted memory B cell populations able to bind monomeric gp70 to express and test anti-env mAbs. One clone termed 1E4 with IgG2c isotype showed the strongest apparent affinity to B16F10 cells at a binding constant ($K_d$) of around 25 nM (S7 Fig). We then created a B16F10 cell line with the *env* knocked down by transient expression of CRISPR-Cas9 (B16F10 envKD) and negative selection with 1E4. Strikingly, all of the sera that showed reactivity to B16F10 did not show any binding to B16F10 envKD cells (Fig 3C, S8 Fig). eMLV env has been previously identified as a tumor antigen [33–35], containing T-cell antigens able to induce protective immune responses [36–38]. One of the peptide antigens found in the transmembrane region of eMLV, p15E, was recently found to be immunodominant in mice bearing MC38 tumors with the use of a knockout cell line [39]. Similarly, by constructing the envKD cell line, we conclusively demonstrated that eMLV env is the primary cell-surface antigen recognized by iiAbs.

## Anti-env mAb confers intravenous challenge protection

We next explored whether anti-env mAbs can protect mice from intravenous tumor challenge. As expected, 1E4 showed reactivity to other tumor cell lines tested, including MC38, EL.4, CT26, and 4T1; however it did not bind EMT6, KP2677, and TC-1 (Fig 4A). Based on our

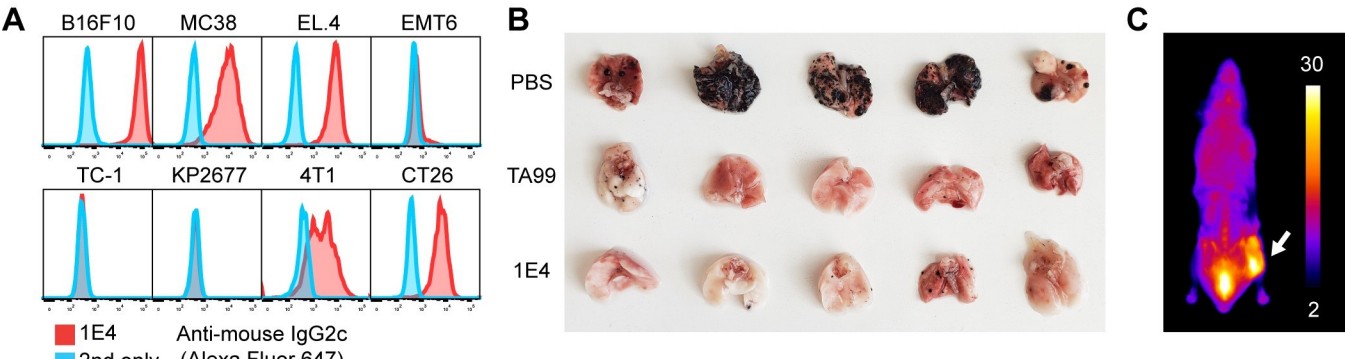

**Fig 4. Anti-env Ab is cross-reactive and protective against intravenous B16F10 challenge.** (A) Indicated cells were incubated with 10 μg/ml 1E4 and analyzed as in Fig 1A. (B) TA99 or 1E4 was injected i.p. into naive mice 6 h before intravenous challenge of $2.5 \times 10^5$ B16F10 cells. Shown are the images of lungs isolated 17 d after the challenge. (C) Five hundred micrograms of Alexa-Fluor-750 labeled 1E4 was intravenously injected into a mouse bearing a B16F10 tumor. Shown is a representative cryo-fluorescence tomography image 48 h after the injection.

finding that the iiAbs in AIPV-treated mice primarily target gp70, we tested whether 1E4 is protective. Similarly to serum from AIPV-treated mice, 1E4 was able to protect naive mice against intravenous B16F10 challenge (Fig 4B), indicating that anti-env Abs in AIPV-treated mice would be capable of contributing to protection [20]. To determine whether 1E4 was specifically targeting tumor cells, we intravenously injected fluorescently-labeled 1E4 into a mouse bearing a B16F10 tumor. Two days later, the mouse was sacrificed and the whole animal was imaged by cryo-fluorescence tomography. We detected a strong signal in the tumor, indicating that 1E4 effectively targeted tumor cells (Fig 4C). To further improve the function of 1E4, the single-chain variable domain (scFv) of 1E4 was affinity-matured by yeast surface display using monomeric gp70 as the antigen. While there was a marked improvement in the affinity of scFv against gp70, there was only a modest improvement in the apparent affinity of mAb to the cell-surface antigen (S7 and S9 Figs). This is likely due to the inability of the monomeric gp70 to recapitulate the conformational epitopes present on the native env trimer. Regardless, we used the affinity-matured clone termed 1E4.2.1 with an affinity at 7.0 nM in subsequent experiments.

### Pre-existing anti-env Ab protects against subcutaneous tumor challenge

As seen previously, anti-tumor Ab treatment alone is ineffective against established B16F10 tumors [20]. However, given the observed efficacy in the intravenous tumor challenge, we investigated whether pre-existing anti-env Abs can reject more challenging subcutaneous tumors. To test this, starting one day before subcutaneous inoculation of $10^5$ B16F10 cells (inoculum of a secondary subcutaneous challenge), we administered five doses of 1E4.2.1 every three days. We observed that two out of five mice were completely tumor-free (Fig 5A), indicating that circulating 1E4.2.1 was capable of clearing the inoculum before it was able to implant. We suspected that the infusion of anti-env mAb may not fully mimic anti-env responses that are likely polyclonal in AIPV-treated mice. Therefore, we vaccinated mice with the RBD protein and adjuvant to induce a polyclonal anti-env Ab response. After three boosts, we observed that the serum Abs of vaccinated mice recognized B16F10 as well as MC38 cells by flow cytometry (S10 Fig). After one additional boost, vaccinated mice were subcutaneously challenged with $10^5$ B16F10 cells. We observed that four out of five mice were tumor-free (Fig 5B) and the one vaccinated mouse that succumbed to tumor burden had a low level of anti-env Abs against B16F10 cells (S9 Fig). This prophylactic vaccine study indicates that a strong anti-env Ab response alone can be sufficient to protect mice from a subcutaneous tumor challenge.

### Discussion

To date, strategies to improve cancer immunotherapies have largely focused on T cells [2, 40]; however, there is a growing recognition that B cells contribute to anti-tumor immunity [41, 42]. To better understand an effective antitumor humoral response, we characterized anti-tumor Abs found to be protective in mice cured by immunotherapy. We observed that the iiAbs in cured mice cross-react with heterologous tumors and are predominantly of IgG2b and IgG2c, the major isotypes that mediate ADCC [29, 43]. Although the original AIPV therapy targeted TRP1 and TRP2, we found that the humoral response spread to primarily one secondary tumor antigen, eMLV env. This humoral response was found to be protective, as anti-env Abs lead to protection following intravenous challenge as well as subcutaneous challenge. Together, this suggests that a strong humoral response alone can mediate tumor control, at least in the early stages of tumorigenesis.

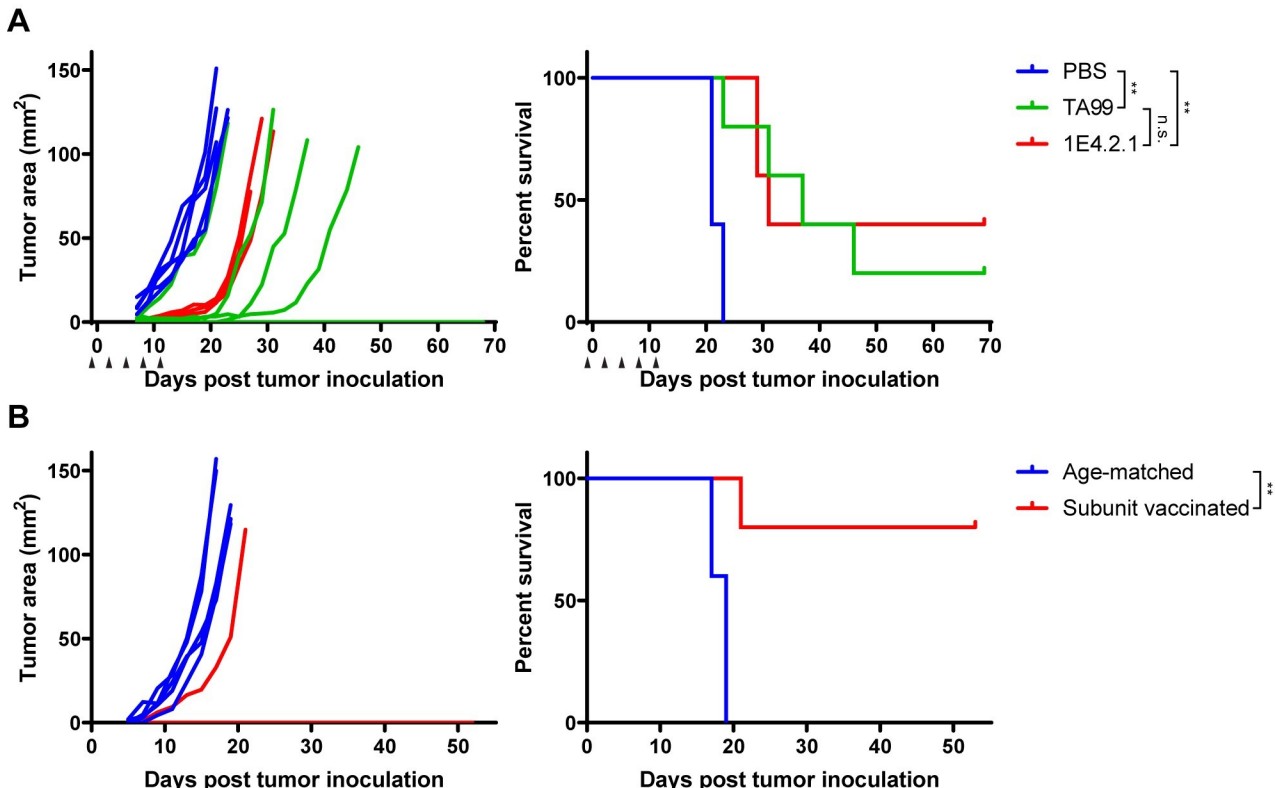

**Fig 5. Anti-env Ab response is protective against subcutaneous B16F10 challenge.** (A) Naive mice were inoculated with $10^5$ B16F10 cells on day 0 and treated with 200 μg TA99 (n = 5), 1E4.2.1 (n = 5), or PBS (n = 5) on days -1, 2, 5, 8, 11. Arrows indicate treatment points. Shown are tumor area curves and overall survival. (B) Age-matched mice (n = 5) and naive mice vaccinated five times with 10 μg RBD and 5 μg saponin-MPLA nanoparticles (n = 5) were challenged with $10^5$ B16F10 cells. Shown are tumor area curves and overall survival. $^*P < 0.05$ and $^{**}P < 0.01$.

Our finding has potential implications for human cancer, as around 8% of the human genome consists of ERV genes [44]. ERV expression is known to be suppressed by epigenetic mechanisms in healthy cells [45], but has been shown to be reactivated in tumor cells [46, 47]. Just as eMLV is expressed by different mouse cell lines, human ERV-K (HERV-K) is expressed by numerous human tumors [48]. However, a key distinction between human and murine ERVs is that eMLV can reconstitute infectious virions [49] that can negatively impact the murine immune system [49, 50], suggesting that successful immunotherapies must mount both antitumor and antiviral immune responses. By contrast, there has been no report of replication-competent HERV in humans [51], suggesting that cell-surface env of HERV-K may be a bona fide therapeutic target. Thus far, gene silencing [52], Ab therapy [53], and CAR-T therapy [54, 55] against HERV-K have shown modest effects in tumor control in preclinical studies. However, these experiments were conducted in immunocompromised mice, thereby not incorporating critical effector-function, tumor microenvironment repolarization, and immune-complex driven enhancement of antigen cross presentation, all shown in many previous studies to significantly amplify self-vaccinal T cell responses [56].

Our work also exposes yet another limitation of mouse models, since mouse strains commonly used for cancer research encode for a particularly immunodominant antigen. We observed that humoral response against eMLV env is predominant in curative immunotherapies and T-cell response against eMLV env was shown to be similarly immunodominant in mouse models [39, 57]. Although convergent humoral responses have been observed in

humans [58], they are against multiple tumor antigens, in stark contrast to the focused humoral response we observed in mice. This discrepancy suggests that mouse strains widely used are inadequately modeling antigen spreading in humans and the scientific community may benefit from using mice knocked out of *Emv2*, the gene element encoding eMLV, and their concomitant syngeneic cell lines.

Recently, human Abs with cross-reactivity to non-autologous tumors and ability to control syngeneic mouse tumor cells have been identified from non-progressing cancer patients [58]. Although the identities of all shared antigens have not been made public, this study parallels our observation of cross-reactive Abs with protective efficacy. These findings call for further serological studies to better understand humoral responses in cancer patients in remission. Our work (in particular the protective vaccination result in Fig 5B) also raises the provocative question as to whether vaccination against shared antigens such as env could lead to cancer prophylaxis.

## Supporting information

**S1 Fig. Anti-tumor iiAbs are predominantly IgG2b and IgG2c isotypes.** B16F10 and MC38 cells were incubated with indicated sera, stained with Alexa-Fluor-647-conjugated monoclonal Abs against different mouse IgG isotype, and analyzed by flow cytometry.
(TIF)

**S2 Fig. Identification of gag by modified 2D immunoblotting.** (A) MC38 membrane proteins and biotinylated BSA, soybean trypsin inhibitor, and equine hemoglobin were immunoblotted with CM7 serum and detected with IRDye-800CW-conjugated anti-mouse IgG2b and IgG2c Abs and Alexa-Fluor-647-conjugated streptavidin. (B) Identical sample in (A) was partially transferred and chromogenically detected with biotinylated anti-mouse IgG2b and IgG2c Abs and HRP-conjugated streptavidin. (C) The gel remaining after the partial transfer was silver-stained and overlaid on the chromogenic immunoblot. Red boxes indicate the spots of interest.
(TIF)

**S3 Fig. Identification of env by modified 2D immunoblotting.** (A) B16F10 membrane proteins were immunoblotted with CM6 serum and detected with IRDye-800CW-conjugated anti-mouse IgG2b and IgG2c Abs. (B) Identical sample in (A) was partially transferred and chromogenically detected with biotinylated anti-mouse IgG2b and IgG2c Abs and HRP-conjugated streptavidin. (C) The gel remaining after the partial transfer was silver-stained and overlaid on the chromogenic immunoblot. Red boxes indicate the spots of interest.
(TIF)

**S4 Fig. All of antisera reactive to B16F10 are competed by gp70.** B16F10 cells were incubated with 1% serum or 10 μg/ml TA99 in the presence of 50 μg/ml gp70 and analyzed as in Fig 3B.
(TIF)

**S5 Fig. Other curative immunotherapies induce anti-env Ab response.** (A,B) B16F10 cells were incubated with 1% serum collected from mice that received indicated immunotherapies in the presence of 50 μg/ml gp70 and analyzed as in Fig 3B.
(TIF)

**S6 Fig. RBD contains major epitopes.** B16F10 cells were incubated with 1% serum in the presence of 50 μg/ml gp70 or RBD and analyzed as in Fig 3B.
(TIF)

**S7 Fig. 1E4 binds to B16F10 at a nanomolar affinity.** B16F10 cells were incubated with soluble 1E4 and affinity-matured clones at indicated concentrations and analyzed as in Fig 1A.
(TIF)

**S8 Fig. None of the antisera bind to B16F10 envKO cells.** B16F10 or B16F10 envKO cells were incubated with 1% serum and analyzed as in Fig 1A.
(TIF)

**S9 Fig. Affinity maturation of 1E4.** Yeast cells expressing scFv's of 1E4 and affinity-matured clones were were incubated with chicken anti-c-myc Ab and biotinylated gp70 at indicated concentrations, washed, stained with Alexa-Fluor-488-conjugated streptavidin and Alexa-Fluor-647-conjugated anti-chicken IgG secondary Ab, and analyzed by flow cytometry.
(TIF)

**S10 Fig. Subunit vaccination with RBD induces anti-env Abs able to bind B16F10 and MC38.** Naive mice were vaccinated on weeks 0, 3, 6, 9, and 12 with RBD adjuvanted with saponin-MPLA nanoparticles (R# for each RBD-vaccinated mouse) and serum was collected on indicated weeks. B16F10 and MC38 cells were incubated with 1% serum, stained with Alexa-Fluor-647-conjugated anti-mouse IgG or IgG2b and IgG2c, and analyzed by flow cytometry. *Asterisk denotes the mouse that succumbed to the tumor challenge.
(TIF)

**S1 Table. Search results of the peptides identified in Fig 2D and S2C Fig. Data was searched against the (A) RefSeq *Mus musculus* database and (B) SwissProt/UniProt database. Orange and b** lue colors indicate independent experiments. Highlighted row indicates the antigen identified.
(TIF)

**S2 Table. Search results of the peptides identified in S3C Fig.** Data was searched against the (A) RefSeq *Mus musculus* database and (B) SwissProt/UniProt database. Orange and blue colors indicate independent experiments. Highlighted row indicates the antigen identified.
(TIF)

**S1 Raw images. Raw immunoblot and gel images files.**
(PDF)

## Acknowledgments

We thank the Koch Institute's Swanson Biotechnology Center for technical support. We thank N. Mehta for assistance in method development. We thank M. Burger for assistance in generating the knockdown cell line. We thank E. Spooner for providing protein mass spectrometry services. We thank M. Farhoud at Emit Imaging for performing cryo-fluorescence tomography.

## Author Contributions

**Conceptualization:** Byong H. Kang, K. Dane Wittrup.

**Funding acquisition:** K. Dane Wittrup.

**Investigation:** Byong H. Kang, Noor Momin, Kelly D. Moynihan, Murillo Silva, Yingzhong Li, Darrell J. Irvine.

**Methodology:** Byong H. Kang.

**Project administration:** K. Dane Wittrup.

**Supervision:** K. Dane Wittrup.

**Writing – original draft:** Byong H. Kang.

**Writing – review & editing:** Byong H. Kang, K. Dane Wittrup.

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
