## [Decision Letter · Decision Letter 0]

6 Jan 2021

Pécs, Hungary

January 5, 2021

PONE-D-20-33132

Immunotherapy-induced antibodies to endogenous retroviral envelope glycoprotein confer tumor protection in mice

PLOS ONE

Dear Dr. Wittrup,

Thank you for submitting your manuscript to PLOS ONE. After careful consideration, we feel that it has merit but does not fully meet PLOS ONE’s publication criteria as it currently stands. Therefore, we invite you to submit a revised version of the manuscript that addresses the points raised by the Reviewers, listed below.

We look forward to receiving your revised manuscript.

Kind regards,

Joseph Najbauer, Ph.D.

Academic Editor

PLOS ONE

Journal Requirements:

2.PLOS ONE now requires that authors provide the original uncropped and unadjusted images underlying all blot or gel results reported in a submission’s figures or Supporting Information files. This policy and the journal’s other requirements for blot/gel reporting and figure preparation are described in detail at https://journals.plos.org/plosone/s/figures#loc-blot-and-gel-reporting-requirements and https://journals.plos.org/plosone/s/figures#loc-preparing-figures-from-image-files. When you submit your revised manuscript, please ensure that your figures adhere fully to these guidelines and provide the original underlying images for all blot or gel data reported in your submission. See the following link for instructions on providing the original image data: https://journals.plos.org/plosone/s/figures#loc-original-images-for-blots-and-gels.

Additional Editor Comments:

Introduction

and immunosurveillance[1,2].

Please chasnge to

and immunosurveillance [1,2].

Please insert space before the reference numbers throughout the manuscript.

Reviewers' comments:

Reviewer's Responses to Questions

**Comments to the Author**

1. Is the manuscript technically sound, and do the data support the conclusions?

Reviewer #1: Yes

Reviewer #2: Yes

2. Has the statistical analysis been performed appropriately and rigorously? 

Reviewer #1: Yes

Reviewer #2: Yes

3. Have the authors made all data underlying the findings in their manuscript fully available?

Reviewer #1: Yes

Reviewer #2: Yes

4. Is the manuscript presented in an intelligible fashion and written in standard English?

Reviewer #1: Yes

Reviewer #2: Yes

5. Review Comments to the Author

Reviewer #1: Kang et al. describe the identification of the mouse tumor antigen underlying a humoral response and tumor challenge response, Env glycoprotein. The study generally was performed well with appropriate rigor to guide the conclusions. The paper will be of interest to others in the oncology field and the discussion does an excellent job capturing the impact (although not key to Plos One) of the article. I have a few minor concerns that should be addressed to help improve the manuscript prior to publication.

Flow figures are completely lacking quantification- although this allows qualitative claims, authors mention things like "we conclusively demonstrated that eMLV env is the primary cell-surface antigen...." the manuscript would help with quantitation to support such statements.

Figure 1 legend gives statistical confidence ranges ** and *** that were not used in the figure as far as I could tell.

Figure 1D- were all 10 CM samples from cured mice that were successful in rejecting the secondary challenge? same or different mice that those 5 shown in 1C?

Figure 2B- indicate the colors found on the blot in the legend that correspond to IR800 and AF-647.

Env and GP70 are used somewhat interchangeably (referring to protein and gene) and could create confusion, moreover later in the paper monomeric versus trimeric possibilities are introduced. The authors should stick with one naming convention throughout the text and figures. They should also be clear about what protein (monomeric GP70?) is being used for flow assays, competition, affinity maturation and vaccination.

Reviewer #2: In this study, Kang et al studied the roles of antibodies against glycoproteins from murine ERV in immunotherapy and discovered their key roles in mediating anti-tumor effect during cancer immunotherapy. The study was based on previous studies in the Witrrup lab showing that in mice cured of their tumors by different immunotherapeutic methods, antibodies against tumor cell surface antigens played critical roles. The authors adopted a systematic approach and was able to identify eMLV gp70 as the commons murine tumor antigen that was shared among different tumor lines. Of particular interest is the finding that antibodies against this antigen had potent anti-tumor effect against multiple syngeneic or allogenic tumor cell line. The study itself is well designed and carried out in a very competent manner. More importantly, it has significant implications for human cancer treatment since ERV expression has been observed in many human tumors as well.

6. PLOS authors have the option to publish the peer review history of their article (what does this mean?). If published, this will include your full peer review and any attached files.

Reviewer #1: No

Reviewer #2: No

---

## [Author Response · Author response to Decision Letter 0]

2 Feb 2021

 The manuscript has been revised to meet the style requirements.

2.PLOS ONE now requires that authors provide the original uncropped and unadjusted images underlying all blot or gel results reported in a submission’s figures or Supporting Information files. This policy and the journal’s other requirements for blot/gel reporting and figure preparation are described in detail at https://journals.plos.org/plosone/s/figures#loc-blot-and-gel-reporting-requirements and https://journals.plos.org/plosone/s/figures#loc-preparing-figures-from-image-files. When you submit your revised manuscript, please ensure that your figures adhere fully to these guidelines and provide the original underlying images for all blot or gel data reported in your submission. See the following link for instructions on providing the original image data: https://journals.plos.org/plosone/s/figures#loc-original-images-for-blots-and-gels.

 The original images are compiled in the S1_raw_images.pdf file. They are for Figs 1D, 2B, 2C, 2D, 3A, S2 and S3.

 The phrase was removed from the manuscript.

 Additional Editor Comments:

Introduction

and immunosurveillance[1,2].

Please chasnge to

and immunosurveillance [1,2].

Please insert space before the reference numbers throughout the manuscript.

Revised.

   Reviewers' comments:  Reviewer's Responses to Questions

Comments to the Author  1. Is the manuscript technically sound, and do the data support the conclusions?  The manuscript must describe a technically sound piece of scientific research with data that supports the conclusions. Experiments must have been conducted rigorously, with appropriate controls, replication, and sample sizes. The conclusions must be drawn appropriately based on the data presented.

Reviewer #1: Yes

Reviewer #2: Yes

2. Has the statistical analysis been performed appropriately and rigorously?

Reviewer #1: Yes

Reviewer #2: Yes

3. Have the authors made all data underlying the findings in their manuscript fully available?  The PLOS Data policy requires authors to make all data underlying the findings described in their manuscript fully available without restriction, with rare exception (please refer to the Data Availability Statement in the manuscript PDF file). The data should be provided as part of the manuscript or its supporting information, or deposited to a public repository. For example, in addition to summary statistics, the data points behind means, medians and variance measures should be available. If there are restrictions on publicly sharing data—e.g. participant privacy or use of data from a third party—those must be specified.

Reviewer #1: Yes

Reviewer #2: Yes

4. Is the manuscript presented in an intelligible fashion and written in standard English?  PLOS ONE does not copyedit accepted manuscripts, so the language in submitted articles must be clear, correct, and unambiguous. Any typographical or grammatical errors should be corrected at revision, so please note any specific errors here.

Reviewer #1: Yes

Reviewer #2: Yes

5. Review Comments to the Author  Please use the space provided to explain your answers to the questions above. You may also include additional comments for the author, including concerns about dual publication, research ethics, or publication ethics. (Please upload your review as an attachment if it exceeds 20,000 characters)

Reviewer #1: Kang et al. describe the identification of the mouse tumor antigen underlying a humoral response and tumor challenge response, Env glycoprotein. The study generally was performed well with appropriate rigor to guide the conclusions. The paper will be of interest to others in the oncology field and the discussion does an excellent job capturing the impact (although not key to Plos One) of the article. I have a few minor concerns that should be addressed to help improve the manuscript prior to publication.  Flow figures are completely lacking quantification- although this allows qualitative claims, authors mention things like "we conclusively demonstrated that eMLV env is the primary cell-surface antigen...." the manuscript would help with quantitation to support such statements.  A new panel (Fig 3D) was added to quantitate the significant change in MFI of B16F10 WT vs envKD cells stained with the antisera.  Figure 1 legend gives statistical confidence ranges ** and *** that were not used in the figure as far as I could tell.  Unused ones were removed.  Figure 1D- were all 10 CM samples from cured mice that were successful in rejecting the secondary challenge? same or different mice that those 5 shown in 1C?

All CM mice had rejected a secondary challenge. This was clarified in the text by changing "sera from cured mice" to "sera from cured mice that rejected a secondary challenge."

Figure 2B- indicate the colors found on the blot in the legend that correspond to IR800 and AF-647.

The legend was revised to indicate the colors.  Env and GP70 are used somewhat interchangeably (referring to protein and gene) and could create confusion, moreover later in the paper monomeric versus trimeric possibilities are introduced. The authors should stick with one naming convention throughout the text and figures. They should also be clear about what protein (monomeric GP70?) is being used for flow assays, competition, affinity maturation and vaccination.

 We intended italicized env to refer to the gene and non-italicized env to refer to the envelope glycoprotein made up of gp70 and p15E subunit proteins. The text has been revised to show gp70 only when solubly expressed gp70 was used in the experiment and where applicable, monomeric nature of gp70 was explicitly stated to avoid confusion.

Reviewer #2: In this study, Kang et al studied the roles of antibodies against glycoproteins from murine ERV in immunotherapy and discovered their key roles in mediating anti-tumor effect during cancer immunotherapy. The study was based on previous studies in the Witrrup lab showing that in mice cured of their tumors by different immunotherapeutic methods, antibodies against tumor cell surface antigens played critical roles. The authors adopted a systematic approach and was able to identify eMLV gp70 as the commons murine tumor antigen that was shared among different tumor lines. Of particular interest is the finding that antibodies against this antigen had potent anti-tumor effect against multiple syngeneic or allogenic tumor cell line. The study itself is well designed and carried out in a very competent manner. More importantly, it has significant implications for human cancer treatment since ERV expression has been observed in many human tumors as well.

6. PLOS authors have the option to publish the peer review history of their article (what does this mean?). If published, this will include your full peer review and any attached files.   Do you want your identity to be public for this peer review? For information about this choice, including consent withdrawal, please see our Privacy Policy.

Reviewer #1: No

Reviewer #2: No

 While revising your submission, please upload your figure files to the Preflight Analysis and Conversion Engine (PACE) digital diagnostic tool, https://pacev2.apexcovantage.com/. PACE helps ensure that figures meet PLOS requirements. To use PACE, you must first register as a user. Registration is free. Then, login and navigate to the UPLOAD tab, where you will find detailed instructions on how to use the tool. If you encounter any issues or have any questions when using PACE, please email PLOS at figures@plos.org. Please note that Supporting Information files do not need this step.

---

## [Decision Letter · Decision Letter 1]

8 Mar 2021

Pécs, Hungary

March 7, 2021

Immunotherapy-induced antibodies to endogenous retroviral envelope glycoprotein confer tumor protection in mice

PONE-D-20-33132R1

Dear Dr. Wittrup,

We’re pleased to inform you that your manuscript (R1 version) has been judged scientifically suitable for publication and will be formally accepted for publication once it meets all outstanding technical requirements.

Kind regards,

Joseph Najbauer, Ph.D.

Academic Editor

PLOS ONE

**DEAR EDITORIAL OFFICE**,

As requested by Dr. K. Dane Wittrup in his Cover letter dated January 23, 2021, please put the following in the funding statement: “This work was supported in part by the Ragon Institute of MGH, MIT and Harvard, the Koch Institute Frontier Research Program, the Kathy and Curt Marble Cancer Research Fund, and the Mayo Clinic – Koch Institute Cancer Solutions Team Grant funding.”

Thank you, and kind regards,

Joseph Najbauer, Ph.D.

Academic Editor

PLOS ONE

Reviewers' comments:

Reviewer's Responses to Questions

**Comments to the Author**

1. If the authors have adequately addressed your comments raised in a previous round of review and you feel that this manuscript is now acceptable for publication, you may indicate that here to bypass the “Comments to the Author” section, enter your conflict of interest statement in the “Confidential to Editor” section, and submit your "Accept" recommendation.

Reviewer #1: All comments have been addressed

2. Is the manuscript technically sound, and do the data support the conclusions?

Reviewer #1: Yes

3. Has the statistical analysis been performed appropriately and rigorously? 

Reviewer #1: Yes

4. Have the authors made all data underlying the findings in their manuscript fully available?

Reviewer #1: Yes

5. Is the manuscript presented in an intelligible fashion and written in standard English?

Reviewer #1: Yes

6. Review Comments to the Author

Reviewer #1: (No Response)

7. PLOS authors have the option to publish the peer review history of their article (what does this mean?). If published, this will include your full peer review and any attached files.

Reviewer #1: No

---

## [Editor Report · Acceptance letter]

6 Apr 2021

PONE-D-20-33132R1 

Immunotherapy-induced antibodies to endogenous retroviral envelope glycoprotein confer tumor protection in mice 

Dear Dr. Wittrup:

I'm pleased to inform you that your manuscript has been deemed suitable for publication in PLOS ONE. Congratulations! Your manuscript is now with our production department. 

Kind regards, 

on behalf of

Dr. Joseph Najbauer 

Academic Editor

PLOS ONE